# Construction of Riboswitches for Screening Antibacterial Agents from Forest Plants

Zhanjun Liu, Taotao Li *, Xingyu Zhang, Shiquan Liu, Zhiyuan Hu, Songlin Yu and Xiaohong Zhou

Hunan Provincial Key Lab of Dark Tea and Jin-hua, Hunan City University, Yiyang 413000, China; liuzhanjun@hncu.edu.cn (Z.L.)
* Correspondence: litaotao@hncu.edu.cn; Tel.: +0737-6353033

**Abstract:** Forest plants contain abundant natural products, providing a valuable resource for obtaining compounds with various functional activities, such as antimicrobial, lipid-lowering, and immunoregulatory activities. The development of efficient tools for rapidly screening functional natural products from forest plants is essential for human health. In this study, we constructed some transgenic strains (*Escherichia coli*) containing Ahy1-1 riboswitches that respond to cyclic di-guanylate (c-di-GMP), serving as a novel bacteriostatic target. The Ahy1-1 riboswitches contained the *LacZ* gene (encoding β-galactosidase) and c-di-GMP aptamer in order to monitor β-galactosidase activity due to changes in c-di-GMP. After co-incubating with extracts from fresh orange peel, fresh tea leaves, and Fuzhuan brick tea, the orange peel exhibited a significant inhibition of c-di-GMP generation. The extract of tea leaves had a minor influence on the synthesis of c-di-GMP, whereas Fuzhuan brick tea, which is fermented by various microorganisms, inhibited the production of c-di-GMP. Our constructed transgenic strains could be used to screen for antibacterial agents from forest plants. Beyond antibacterial agents, other functional compounds from forest plants could be selected by designing diverse riboswitches.

**Keywords:** plant bioactive components; antibacterial target; non-coding RNA; antibiotic resistance; functional nucleic acid material





## 1. Introduction

Forests are a significant economic resource in China [1–3], containing rich and widely distributed resources. By 2023, China's forest coverage rate will have reached 24.02%, with 220 million hectares of forest area and 17.76 billion $m^3$ of forest stock. These forest resources include a rich variety of tea (*Camellia sinensis* L.) tree species. According to their growth forms and characteristics, tea plants can be divided into arbor-type, small arbor-type, and shrub-type. The arbor-type tea tree varieties are mainly distributed in the south of China, including Guangdong, Guangxi, Fujian, and Hainan Provinces. The small arbor-type tea tree varieties are relatively concentrated in Fujian, Guangdong, Guangxi, Hunan, and Jiangxi Provinces. Overall, China has a vast area of tea plantations, accounting for about 30% of the world's total tea area. On the other hand, China is also renowned as a treasury of orange resources in the world, with a wide variety of orange species, including *Citrus reticulata Blanco*, *Citrus sinensis* (L.), *Citrus tangerina Tanaka*, and so on. The Yangtze River Valley is the largest citrus production region in China, covering an area of more than 200,000 hectares. The main cultivars include oranges, tangerines, and pomelos. This region has an average annual temperature of 17.5–18.5 °C and receives about 1300 mm of rain a year. A large number of bioactive compounds with various physiological activities, such as antioxidant activity (catechins and tea polysaccharides extracted from tea), antibacterial activity (flavonoids extracted from orange peel and tea leaves) [4–6], antiviral activity (flavonoids and polyphenols extracted from *Isatis tinctoria*), antineoplastic activity (paclitaxel extracted from *Taxus* L.), immunoregulatory activity (ginsenoside extracted from *Panax ginseng* C. A. Mey), antimicrobial activity, lipid-lowering

activity, and so on [7–11], could be acquired from different plants. Bouvardin is one of the bacteriostatic agents produced in plant cells when infected by fungus and pathogenic microorganisms, or when exposed to toxic chemicals and ultraviolet light [12–14]. The antibacterial spectrum of plant extracts is wide, including viruses, fungi, and bacteria, which generally inhibit pathogenic microorganism growth and protect probiotics. Due to the high-worth bacteriostatic agent bouvardin, a novel screening strategy to rapidly obtain various new plant extracts with antimicrobial activity would expand the application of plant forests and enhance plants' economic value.

The emergence of antibiotic resistance In pathogenic bacteria has Increasingly become a significant medical challenge. Therefore, the development of new antibiotics to fight against bacteria is imperative [15]. The germicidal mechanisms of commercial antibiotics include inhibiting cell wall synthesis, changing cell membrane permeability, preventing protein synthesis, inhibiting nucleic acid synthesis, and preventing folic acid synthesis. However, the widespread use of a newly developed antibiotic may lead to a rapid loss of its antimicrobial activity. To solve the antimicrobial resistance problem of antibiotics, some new antibacterial targets for antibiotics should be excavated. A riboswitch is a potential antibacterial target [16]. Riboswitches are cis-acting RNA elements that bind specific small-molecule metabolites and regulate gene expression [17]. Members of riboswitches are discovered in the 5′ untranslated region of messenger RNAs. They are widely distributed in Gram-positive and Gram-negative bacterium metabolite genes and are found in fungi, plants, and human vascular endothelial growth factors [18]. Riboswitches control a broad range of genes in bacterial species related to metabolism, signal conduction, and uptake of amino acids, nucleotides, cofactors, and metal ions [19]. A typical riboswitch contains two domains: an aptamer-binding target and an expression platform. The aptamer domain is highly conserved and forms a unique secondary structure specific to each riboswitch class. The expression platforms typically turn off gene expression in response to the small molecule, while some turn them on [20]. Comparing the highly conserved aptamer domain, the nucleotide sequence and structural configuration of expression platforms are various. This versatility in expression platform permits the riboswitch class to regulate gene expression at several levels via the same targets. The modulation of gene expression could occur by regulating the transcription elongation of mRNA and initiating genetic translation efficiency [21].

Cyclic di-guanosine monophosphate (c-di-GMP) is one of the second messengers in most bacteria. It is a circular RNA dinucleotide and leads to wide-ranging physiological changes in bacteria, including virulence gene expression and cell differentiation [22]. C-di-GMP molecules participate in extensive signal pathways and interact with quorum sensing (QS) [23]. Cellular levels of c-di-GMP are controlled by the opposing activities of diguanylate cyclase (DGC) and phosphodiesterases (PDEs), which harbor the GGDEF domain and EAL domain, respectively [24]. These c-di-GMP riboswitches are bacterial RNA structures that exist upstream of the open reading frames (ORFs) in some organisms and are controlled by c-di-GMP for regulating the expression of its biosynthesis and transport genes. Using genome sequencing analysis, the conserved aptamer of the c-di-GMP riboswitch was considered to consist of at least two stems [25].

Inspired by the natural RNA thermosensor, we constructed some intragenic synthetic riboswitches to detect target chemicals in vivo. Motivated by the importance and widespread function of c-di-GMP as a second messenger, we developed five synthetic riboswitches containing c-di-GMP-specific aptamers, which could control the genetically modified expression of β-lactamase as a response to c-di-GMP and its structural analogs. The c-di-GMP can induce conformational changes in the riboswitches, which results in a reduction or increase in the β-lactamase gene expression, ultimately leading to the diminished or enhanced expression of β-lactamases. To enhance the sensitivity of the synthetic riboswitches, we cloned two metabolite-binding aptamers of the tested functional riboswitches in tandem in front of the expression platform and obtained a switch that functions cooperatively. Following the construction of the riboswitches, we used them to screen

antimicrobial agents and found that orange peel and Fuzhuan brick tea extracts could regulate c-di-GMP biosynthesis. Our constructed transgenic strains with designed riboswitches can be used as engineered bacteria for the rapid screening of bacteriostatic substances.

## 2. Materials and Methods

### 2.1. Materials, Strains, and Culture Conditions

All the molecular biology reagents were of analytical grade and purchased from Shanghai Sangon Biotech Co., Ltd. (Shanghai, China). Antibiotics were purchased from Sigma-Aldrich (Saint Louis, MO, USA). The C4-HSL was synthesized in the Synthetic Medicinal Chemistry Lab (Ocean University of China, Qingdao, China). DNA oligonucleotides and enzymes were purchased from Sigma-Genosys (Saint Louis, MO, USA). *Aeromonase hydrophila* (*A. hydrophila*) and *Bacillus cereus* (*B. cereus*) were purchased from China General Microbiological Culture Collection Center (Beijing, China). *Vibrio cholerae* (*V. cholerae*) O1 biovar eltor strain N16961 genomic DNA was purchased from Shanghai Sangon Biotech Co., Ltd. (Shanghai, China). All plasmid vectors, including pET-24a and pACYC177, were purchased from Thermo Scientific Limited Liability Company (Waltham, MA, USA). Competent DH5-alpha *Escherichia coli* (*E. coli*) was found in our Biolabs (Ocean University of China, Qingdao, China). The β-galactosidase assays kit was purchased from Shanghai Sangon Biotech Co., Ltd. (Shanghai, China).

Transgenic strains were cultured aerobically to the exponential phase in either Luria-Bertani (LB) medium or kanamycin (50 μg/mL) at 37 °C under continuous shaking (150 rpm).

### 2.2. Plasmid Construction

The 5′ untranslated regions (UTRs) upstream of putative open reading frames (ORFs) in *V. cholerae*, *A. hydrophila*, and *B. cereus* were amplified using a PCR. The list of primers is presented in Table 1. The amplified UTR was used as XbaI–BamHI fragments and cloned into the plasmid vector pET-24a. The obtained transcriptional fusant contained a promoterless *lacZ* gene or *sacB* gene. The constructed plasmid vector contained the promoter elements upstream of the UTR and the translation initiation region. This translation initiation region could respond to the riboswitch reporter gene.

**Table 1.** Riboswitch PCR primers.

| Gene | Primers (5′ to 3′) |
| --- | --- |
| Ahy1-1 (Forward) | TCTAGACAGTGAGCCAACGCACATTAC |
| Ahy1-1 (Reverse) | GGATCCGGCAGCCAAAGCCACCAGC |
| VC-2 (Forward) | TCTAGAATAACGCCTATATTTGAAAGC |
| VC-2 (Reverse) | GGATCCGTTTAATACTGGTTTATCCATGC |
| VC-1 (Forward) | TCTAGAAAGCGTGAGAGCTTGATTCCA |
| VC-1 (Reverse) | GGATCCGGTCATTTTAGGTTGTTTTTTCA |
| BC-1 (Forward) | TCTAGAATAAATACCCGAAGAAATCC |
| BC-1 (Reverse) | GGATCCGAACGATAACTTATGCCAATA |
| BC-2 (Forward) | TCTAGAACTAAGCCCCGAGTTAAGAG |
| BC-2 (Reverse) | GGATCCGCTTTTAGTACTTTTCATTTGC |
| *adrA* (Forward) | CATATGTTCCCAAAAATAATGAATGATG |
| *adrA* (Reverse) | GGATCCTCAGGCCGCCACTTCGGTG |
| *LacZ* (Forward) | GGATCCACCATGATTACGGATTCAC |
| *LacZ* (Reverse) | GAATTCTCAGTTGCACCACAGATGAAAC |

To overproduce the *adrA* protein, its biosynthetic genes were extracted from the *E. coli* MG1655 chromosome and amplified using a polymerase chain reaction (PCR). The amplified *adrA* gene was cloned into the region between the XhoI and HindIII sites of a modified pACYC177 vector. This modified vector was constructed using ligation-independent cloning, as described previously [26]. The XhoI and HindIII sites were located downstream of the promoter in the constructed vector. All primers used in this work are

listed in Table 1. Assembled plasmid constructions were verified using sequencing (Thermo Fisher Scientific-CN, Shanghai, China).

### 2.3. Plasmid Transformation

Plasmid transformation was carried out following the manufacturer's directions. The *Escherichia coli* was cultured overnight in LB broth at 37 °C and 120 rpm for 24 h. The cultured *Escherichia coli* cell was rinsed with LB broth three times to reduce the antigen substance at the surface of the cell. The target plasmid was transferred into *Escherichia coli* cells using the heat shock method. The target plasmid (containing $CaCl_2$) was added to the treated *Escherichia coli* solution. The mixture was placed in a water bath (42 °C) for 45 s. After the water bath, the mixture was instantly transferred to ice and remained in these conditions for 5 min. The obtained transplastomic cell lines were screened in a regeneration medium including kanamycin (50 µg/mL).

### 2.4. β-Galactosidase Assays

Before the β-galactosidase assay, the fermentation broth was diluted to an OD600 of 0.1. The obtained colonies were cultured overnight in LB broth containing 50 µg/mL kanamycin at 37 °C and 120 rpm. The β-galactosidase assay was performed using the following standard protocol, when the OD600 of cultures was 1 [27]. A total of 0.25 mL o-nitrophenyl-β-D-galactoside (ONPG, concentration: 0.75 mol/L) was added to the above cultures. The mixture was incubated in a 35 °C water bath for 30 min, and the β-galactosidase activity was measured using an ultraviolet spectrophotometer (UV2600, Shimadzu Corporation, Kyoto, Japan) at 420 nm. The unit of measurement for the activity of the β-galactosidase is 10 U/mg. The β-galactosidase assays of *E. coli* strains containing VC-1, VC-2, BC-1, BC-2, and Ahy1-1 were carried out as described above.

### 2.5. Quantitative Analysis of adrA

*AdrA* transcript levels were determined using real-time quantitative PCR. The reagents, including buffer, primers, sterile water, and Taq DNA polymerase, were added to a PCR tube. The PCR process was conducted as follows: 93 °C for 7 min, followed by 35 cycles of 95 °C for 30 s, 66 °C for 30 s, and 72 °C for 35 s. Pairs of PCR primers are detailed in Table 2. The gene expression levels were normalized against housekeeping control 16S.

**Table 2.** RT-PCR primers.

| Gene | Forward Primers | Reverse Primers |
|------|-----------------|-----------------|
| *adrA* | 5′ AAAACGAACATCAGCGGTCC 3′ | 5′ GCGTTGAAGCAATCGGTAAGA 3′ |
| 16S | 5′ GCGCAACCCTTGTCCTTAGTT 3′ | 5′ TGTCACCGGCAGTCTCCTTAG 3′ |

### 2.6. Screening c-di-GMP Biosynthesis Regulator from Orange Peel and Tea

Fresh orange (*Citrus reticulata Blanco*) was purchased from the Greenery Fruit Co., Ltd. (Yiyang, China). Fresh tea leaves (*Camellia sinensis* L.) and Fuzhuan brick tea (produced by the same tea leaves) were purchased from Hunan Provincial Baishaxi Industry Co. Ltd. (Yiyang, China). The samples were dried at 50 °C for 1 d and crushed with an ultrafine grinder (Zhongde Yishanyuan Pension Service Co., Ltd., Rizhao, China). Then, 10 g samples were added to 100 mL of 50% ethanol and extracted at 50 °C for 4 h. The mixed solutions were centrifuged at 4500 r/min, and the liquid supernatant was collected for screening the c-di-GMP biosynthesis regulator. Then, 10 mL liquid supernatant was added to 90 mL LB medium (containing 2 mL transgenic strains) and incubated at 37 °C under continuous shaking (150 rpm) for 24 h. The β-galactosidase activity of the fermentation broth was measured.

### 2.7. Statistical Analysis

All the samples were selected randomly and experiments were repeated at least three times. Statistical analyses were performed using SPSS (Statistical Product and Service Solutions) 22.0 statistical software (IBM Corp., Armonk, NY, USA). Values were presented as mean ± standard deviation. A one-way analysis of variance was performed to evaluate the results. The level of significance was set at $p < 0.05$.

## 3. Results

### 3.1. Construction of Intragenic Synthetic c-di-GMP Riboswitches

Five riboswitch sequences from *V. cholerae*, *A. hydrophila*, and *B. cereus* were examined for cell function. We inserted the c-di-GMP riboswitch sequences after the start codon of a reporter gene into an "off" switch. A structural change was observed in the 5′ proximal coding region due to the insertion of the c-di-GMP riboswitch (Figure 1A). This changed DNA structure could result in a reduced ribosomal accessibility and protein translation repression.

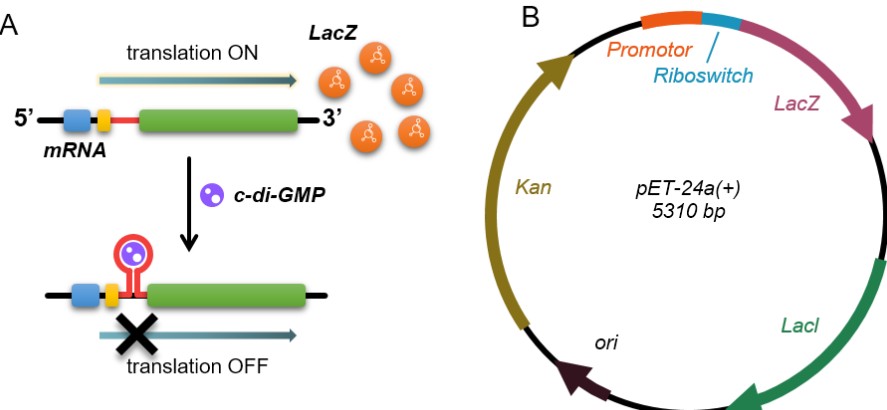

**Figure 1.** Construction of riboswitch responsive to c-di-GMP: (**A**) Schematic diagram of riboswitch switching mechanism. The c-di-GMP aptamer sequence, *lacZ* reporter gene, and start codon were labeled red, green, and yellow, respectively. The mRNA-translated domain was shown in a green arrow. (**B**) The plasmid construction of pET-24a *LacZ* harbors the riboswitch.

The *LacZ* gene was selected as a reporter to evaluate the riboswitch behavior. The construction of an intragenic riboswitch in the plasmid is presented in Figure 1B. The constructed plasmid with a riboswitch has a high copy number in transformed *E. coli* cells. The *E. coli* cells containing aptamer-reporter constructions were cultured in a liquid nutrient medium to detect the riboswitches' functionality in bacteria by assaying for β-galactosidase activity. To determine the functionality of the constructed riboswitches, we utilized the compatible vector pACYC177 to overexpress the *adrA* gene. This gene could encode the DGC protein *adrA* and increase the concentration of c-di-GMP in bacterial cells. The c-di-GMP could promote the synthesis of extracellular polysaccharides via bacteria, so the Congo red plate can be used to detect the activity of the overexpressed *adrA* protein in bacteria. If the overexpressed protein has DGC activity, the colony appears red on the plate, and the darker the red, the stronger the activity. *AdrA* protein-induced expression showed red colonies on the Congo red plate (Figure 2A).

Quantitative real-time PCR indicated that the mRNA levels of *adrA* were upregulated (Figure 2B). Figure 3 illustrates the growth trajectory of *E. coli*. The growth curves of *E. coli* remain largely unchanged upon the introduction of the vector pACYC177, containing *adrA*. This observation suggests that the introduced riboswitch has minimal impact on the growth of these microorganisms. However, no β-galactosidase activity change was detectable with the synthetic c-di-GMP switches (BC-1 and BC-2) from *B. subtilis*, possibly indicating that the riboswitches from the Gram-positive bacteria *B. subtilis* cannot execute a function in the Gram-negative surrogate *E. coli*. The results also indicate that the VC-1 and VC-2 switches

are "on" switches due to a higher gene expression when the concentration of c-di-GMP increased. The Ahy1-1 switch is an "off" switch due to its lower gene expression when the concentration of c-di-GMP increases. The VC-1 and VC-2 switches from *V. cholerae* presented a relatively weak inducibility by c-di-GMP, while the synthetic Ahy1-1 switch from *A. hydrophila* responded much stronger (Figure 4). These results may suggest that the "off" switch is more sensitive than the "on" switch.

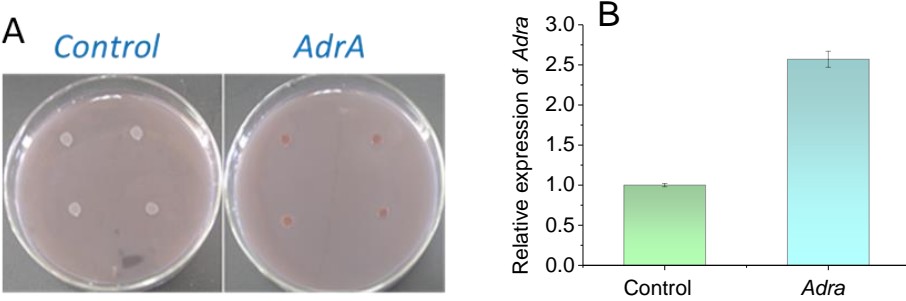

**Figure 2.** (**A**) Functional verification of *adrA*; (**B**) The relative mRNA expression level of *adrA* using real-time quantitative PCR.

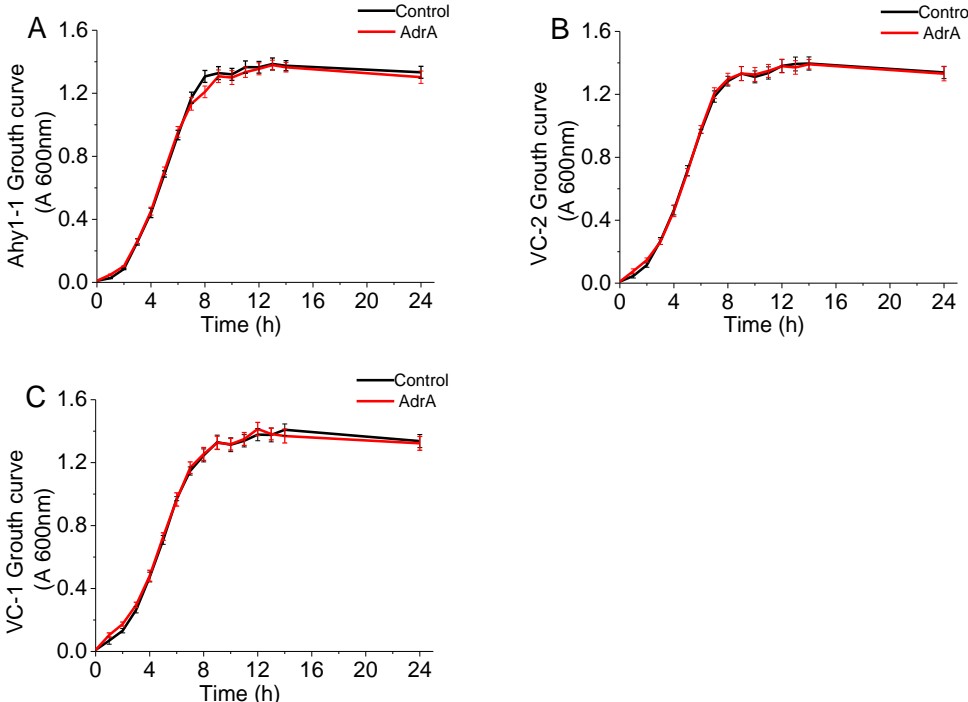

**Figure 3.** Growth curve of *E. coli* cells that contained the (**A**) Ahy1-1, (**B**) VC-2, and (**C**) VC-1 riboswitches containing *adrA* and control plasmid.

Most riboswitches comprise a single micromolecule-binding aptamer and an expression platform. This construction of the gene platform positively responded to the change in metabolite concentrations in the solution. In rare instances, two aptamers of riboswitches approached each other in untranslated regions [28]. Sometimes, two complete riboswitches could also be close to each other. It is reported that an example of a riboswitch in Bacillus anthracis (one of the Gram-positive bacteria) contained two complete glycine riboswitches. These glycine riboswitches included two ligand-binding domains [29]. The two domains cooperated in the recognition of metabolites and closely approximated a two-state genetic switch. This unique form of riboswitch is much more sensitive and responsive to smaller changes in glycine concentration. To this end, we cloned the two metabolite-binding aptamers of the three tested functional riboswitches in tandem before the expression platform

and reporter gene, and then transformed the vector pACYC177 to overexpress *adrA*. By testing β-galactosidase activity, the results indicate that the tandem riboswitch of Ahy1-1 can increase the sensitivity of ligand binding, although the predicted sensitivity of ligand binding was not as distinguished as in a perfect cooperative tandem system, because the tandem riboswitch system was less than the cooperative glycine riboswitch (Figure 5). However, no *lacZ* expression change was detectable with the synthetic vc-1 and vc-2 riboswitches. These results suggest that riboswitches do not increase function in tandem.

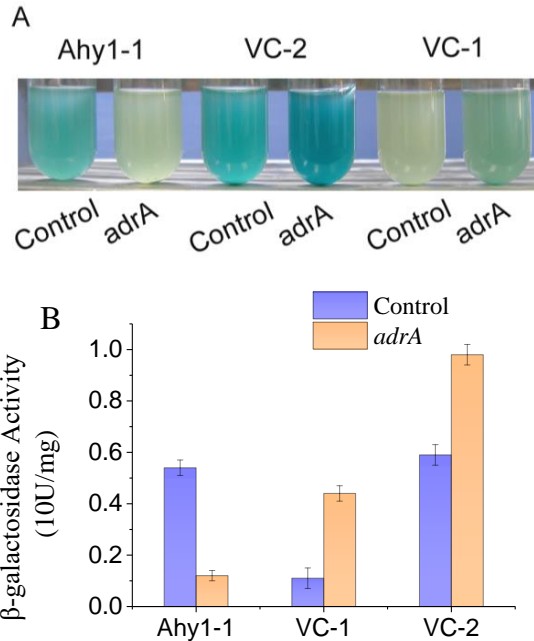

**Figure 4.** The function test results of the Ahy1-1, VC-2, and VC-1 riboswitches: (**A**) Relative expression of reporter gene *lacZ* in *E. coli* cells carrying Ahy1-1, VC-2, and VC-1 riboswitches in glass tubes; (**B**) β-galactosidase activity of *E. coli* cells carrying Ahy1-1, VC-2, and VC-1 riboswitches containing *adrA* and control plasmid.

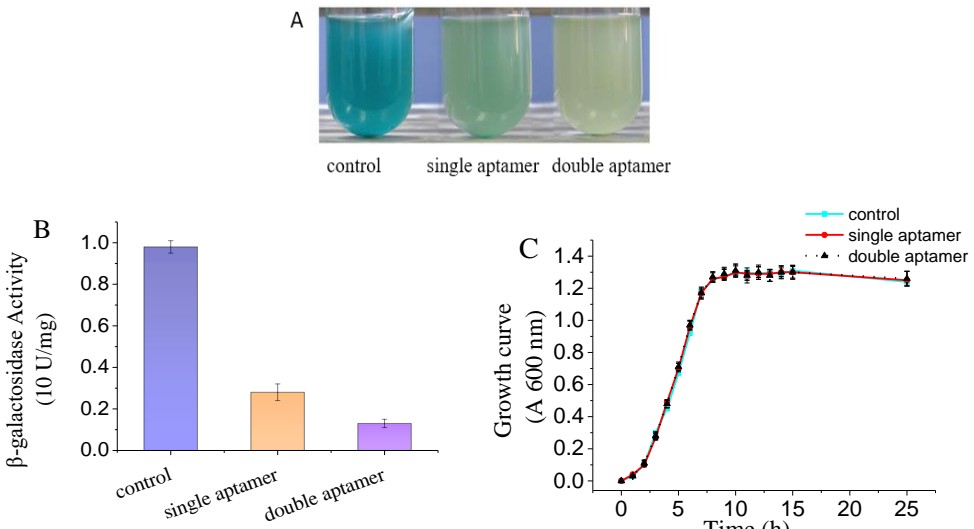

**Figure 5.** The function test results of the Ahy1-1 riboswitch in tandem: (**A**) Relative expression results of reporter gene *lacZ* in *E. coli* cells carrying no, single, and double Ahy1-1 aptamers in glass tubes, respectively; (**B**) β-galactosidase activity; (**C**) Growth curve.

To determine the relationship between QS and c-di-GMP, we added the QS signaling molecule N-butanoyl homoserine lactone (C4-HSL) into the LB broth, whose concentration

did not influence the growth of the surrogate *E. coli*. Overnight, cultures of surrogate *E. coli* transformed, with pET-24a containing Ahy1-1 in tandem being sub-cultured to an OD600 of ~1, and then the β-galactosidase activity was measured. The results showed that the β-galactosidase activity was enhanced when C4-HSL was added (Figure 6). It was believed that GGDEF-containing proteins synthesizing c-di-GMP could be inhibited by C4-HSL. Hence, our results indicated a possible regulatory mechanism for the concentration change of c-di-GMP in the cell.

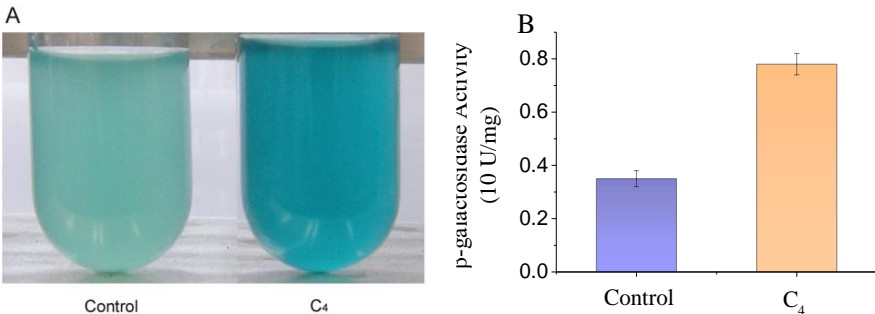

**Figure 6.** The relationship between quorum sensing and c-di-GMP: (**A**) Relative expression of the reporter gene *lacZ* in *E. coli* cells when added to C4-HSL and (**B**) β-galactosidase activity of *E. coli* cells when added to C4-HSL.

### 3.2. Screening c-di-GMP Biosynthesis Regulators in Plants

Low expression of c-di-GMP may slow bacterial growth. To screen the c-di-GMP biosynthesis regulator from forest plants, extracts of orange peel, tea leaves, and Fuzhuan brick tea were added to the constructed transgenic strain solution and incubated for 24 h. As shown in Figure 7, the fermentation broth containing the extracts of orange peel, tea leaves, or Fuzhuan brick tea had different β-galactosidase activity. The orange peel extract decreased β-galactosidase activity in the fermentation broth, indicating the inhibition of c-di-GMP expression in the engineered bacteria. A low concentration of c-di-GMP prevents the bacteria from forming biofilms, which results in bacterial death. Thus, the orange peel may contain some compounds that inhibit bacterial growth due to its lower c-di-GMP expression. The tea leaf extract had a minor influence on β-galactosidase activity. This result indicates that the antibacterial mechanism of the tea leaf extract did not involve regulating c-di-GMP synthesis. However, the Fuzhuan brick tea, which was fermented by various microorganisms, inhibited the production of c-di-GMP by bacteria. These results suggest that orange peel and Fuzhuan brick tea can be used to screen for novel bacteriostatic agents to address antibiotic resistance in pathogenic bacteria, based on the inhibition of c-di-GMP synthesis.

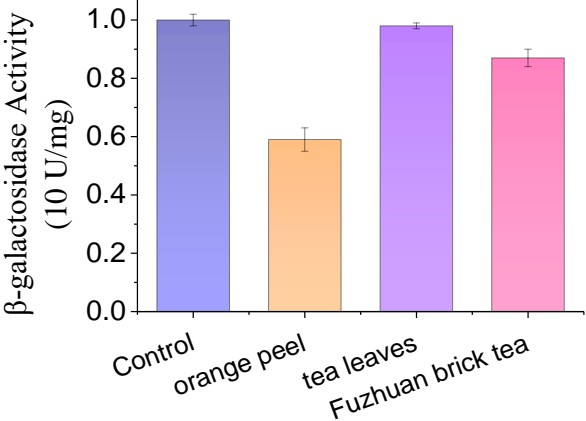

**Figure 7.** β-galactosidase activity in fermentation broth containing an extract of orange peel, tea leaves, or Fuzhuan brick tea, respectively. The constructed transgenic strains contained the VC-2 riboswitch.

## 4. Discussion

Bacterial resistance to antibiotics has been a significant issue in the present society, especially in hospitals where there is an excessive and prolonged use of antibiotics to treat infections. This excessive use of antibiotics prompted some bacteria to obtain multiple drug resistance (MDR). An important cause is the limited number of antibacterial drug targets [30]. Riboswitches have already been identified as a useful antibacterial drug target, offering an effective solution to the fast-growing MDR problem. Because of that, riboswitches have been discovered to highly and selectively respond to small drug-like metabolites [15]. There are many examples of genetic evidence demonstrating that riboswitches would become an excellent target for discovering new antibacterial drugs [31]. Some classes of riboswitches, such as thiamine pyrophosphate (TPP) or flavin mononucleotide (FMN), are discovered in many wild bacterial strains and can be developed as targets for antibacterial drugs. Other riboswitch classes with more sparse distributions would be targeted for new drugs with various functionalities. A metabolite combined with the receptor for riboswitches would inhibit or increase the expression of the reporter gene encoded by the mRNA. Hence, it should be possible to design highly selective ligand analogs that cannot bind to non-target metabolites in bacterial cells, thus shutting down or turning on the expression of a gene particularly regulated by the riboswitch. For drug companies, this strategy could improve their chances of success in finding new antibacterial drugs targeting riboswitches. Based on modern technologies such as rational drug design and synthetic riboswitches, new drugs would be discovered for high-throughput screening. Given the importance of c-di-GMP and c-di-GMP riboswitches, it may be successful to construct new antibacterial chemical compounds binding to c-di-GMP riboswitches and inhibiting the expression of some important genes. In this research, we focused on the development of a simple riboswitch model for predicting the behavior of a switch (i.e., an on- or off-switch). We believe that our designed riboswitch model could be used as a useful tool for drug discovery. The results in Figures 3 and 6 proved that the sensing and modulation abilities of the constructed riboswitch could be used to design sensors for detecting target chemicals in vivo.

The utility of some tandem riboswitch systems is ambiguous and not well established. If the two aptamers have a perfect cooperation, the two closely related aptamers will cooperatively bind to two identical targets and sensitively respond to smaller changes in target concentration due to the gene control element. Our results indicated that the two existing aptamers increased gene control. Three possible characteristics might benefit gene expression regulated by independent tandem riboswitches binding to the same compound. First, the two aptamers would extend the gene expression dynamic range. Second, it has been pointed out that the two independent riboswitches' tandem arrangement will generate a genetic switch to control gene expression under a lower ligand concentration. Thirdly, the inherent characteristic of independently functioning tandem riboswitches was similar to Kd values, and showed an increase in the function of the genetic switch [27].

Owing to its simple construction, the constructed riboswitch could also find broad applications in vivo bioimaging to design new bimolecular tools. Their bioimaging tools could be used to study specific cellular response mechanisms of natural or artificial cellular metabolites by displaying the programmable, complex behavior of chemical cues. The changes in gene expression of bacteria were usually related to their surrounding environment (chemical substances and vicinal bacterial community) and presented by intra- and extra-cellular chemical signals [32]. The detection of this chemical signal information is important to directly monitor the responses of bacteria to fluctuating environmental conditions. QS and c-di-GMP systems are two vital chemical signaling pathways. Extracellular QS and intracellular c-di-GMP signals control some of the same complex processes, such as biofilm formation and virulence [33]. Therefore, there is a relationship between the QS signaling pathways and c-di-GMP signaling pathways. Suppose a molecular link is discovered between QS and c-di-GMP signaling. In such a case, it will verify that bacteria could produce intracellular second messenger signals in the cell due to the change in cell

density. This change in intracellular second messenger would control cellular physiology, gene expression, and group behavior. The results tested by the riboswitch model in Figure 7 demonstrate that the concentration of c-di-GMP in vivo would be repressed by C4-HSL. These results consist of what was reported by G. Kovacikova [34]. In T. maritima, QS could adjust the expression of DGC and PDEA genes and then regulate its encoded protein activities, so QS controls the cellular concentration of c-di-GMP. However, the mechanisms involved in QS and c-di-GMP signaling are not clearly understood to date. Future research needs to determine the internal connection between extracellular QS signaling and intracellular c-di-GMP signaling pathways.

The extract of orange peel demonstrates the capacity to reduce β-galactosidase activity, indicating the presence of compounds that may interact with the riboswitch. The primary antibacterial component found in orange peel is hesperidin [35], which has a spatial configuration conducive to the effective binding with the aptamer section of the riboswitch. In contrast, fresh tea leaf extract has a minimal impact on β-galactosidase activity. However, when processed into Fuzhuan brick tea, the extract's inhibitory effect on the enzyme is significantly enhanced. This phenomenon implies that the fermentation process involved in the production of Fuzhuan brick tea (a fermented dark tea) facilitates the generation of secondary metabolites by microorganisms, which may act as novel antibacterial agents, such as teadenols [36].

## 5. Conclusions

In this study, we constructed an engineered bacteria containing synthetic c-di-GMP riboswitches for discovering new bacteriostatic agents (c-di-GMP synthesis inhibitors) from forest plants. The synthetic c-di-GMP riboswitches were utilized to construct inducible knockout lines for essential genes by simply adding the target gene under the c-di-GMP gene and transferring it into the genome to replace the resident gene. The introduced riboswitch had a minimal impact on the growth of Escherichia coli. The changes in c-di-GMP in the engineered bacteria treated with various plant extracts (orange peel, tea leaves, and Fuzhuan brick tea) were monitored through visualization. The c-di-GMP sensitivity of the engineered bacteria was enhanced with two riboswitches. The extracts of orange peel were found to inhibit c-di-GMP synthesis in bacteria. Compared with fresh tea leaves, Fuzhuan brick tea significantly inhibited c-di-GMP generation. Our engineered bacteria proved to be an efficient tool for rapidly screening c-di-GMP synthesis inhibitors to address the antibiotic resistance in pathogenic bacteria from numerous forest plants.

**Author Contributions:** Conceptualization, T.L. and Z.L.; methodology, Z.L. and S.L.; validation, X.Z. (Xingyu Zhang) and S.Y.; formal analysis, T.L. and X.Z. (Xiaohong Zhou); investigation, Z.H. and T.L.; resources, S.Y. and S.L.; writing—original draft preparation, T.L. and Z.L.; writing—review and editing, Z.H. and Z.L.; visualization, S.L. and T.L.; project administration, T.L. and Z.L.; project administration, T.L. All authors have read and agreed to the published version of the manuscript.

**Funding:** This work was supported by the Research Foundation of the Education Bureau of Hunan Province (22B0785), the Natural Science Foundation of the regional joint fund project of Hunan Province (2023JJ50338), and the Hunan Provincial Key Lab of Dark Tea and Jin-hua (2016TP1022).

**Data Availability Statement:** The data presented in this study are available on request from the corresponding author.

**Conflicts of Interest:** The authors declare no conflicts of interest.

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
