# Peer review of "Construction of Riboswitches for Screening Antibacterial Agents from Forest Plants"

_forests, doi:10.3390/f15020367_

Round 1
Reviewer 1 Report
Comments and Suggestions for Authors
The article entitled ‘Construction of Riboswitches for screening antibacterial agents from forest plants’ shows a new tool for antibacterial compound screening.
The paper has been submitted to the special issue of Forests ‘Advances in the Cultivation, Protection and High-Value Utilization of Forest Resources’.
Maior comments:
The readers of the work will be researchers from all over the world. What is obvious to the authors is not obvious to everyone. So:
In the first paragraph of the Introduction, the authors write about forests in China. And then they smoothly move on to what are supposed to be examples of highly active metabolites of plant origin. Most of them have nothing to do with the forest (e.g. Isatis) and/or China (e.g. Bouvardia). Since this is the main point of connection of the work to the title of the journal as a special issue, it should be corrected. It is really inappropriate to mention tomato or mint here. The selection of species should be considered very carefully. China, as a large country, has a rich flora and you can certainly find native species with valuable properties. A forest is a type of ecosystem and, depending on the geographical latitude has a greater or lesser number of layers and species. However, it is easy to determine whether a given species belongs to such an ecosystem. All names of species should be given in Latin.
1. Lines 93-4 'forest plants............ orange peel and Fuzhuan brick tea extracts'. Citrus × sinensis and Camelia sinensis or are other species the source of these products and where do they grow in nature? What type of forest is there and where is the source of these species?
2. In line 259 the authors state the same. My question is: Is it true? Are they forest species?
3. Subchapter 2.6. Do I understand correctly that the material was bought on the market? Do the authors know anything about the manufacturer? Are they sure of the origins? Are they sure of their species affiliation? From a scientific point of view, this choice of material is not the best.
4. Line 155 ‘Orange tea leaves and Fuzhuan brick tea’ is probably a mistake. Should it be ‘Orange peel, tea leaves, and Fuzhuan brick tea’?
5. Was ethanol evaporation a problem? Was the amount of supernatant the same for each raw material?
6. What statistical analysis was performed using SPSS? only mean and SD? Are the results statistically significant? There are no markers on the graphs.
7. Fig. 3. How was it determined that there was no inhibition? Numerical values, parameters, calculations, or the so-called by-eye, looking at the chart? This should be calculated and quantified.
8. In the discussion, the results should be discussed in the context of the chemical composition of the extracts. If the authors do not have their own results, there are many publications on this topic. The possibility of influence through other mechanisms should be taken into account. See line 265 as well.
Minor comments:
9. The names of the bacteria should be given in full when first used.
10. Line 100-1 What kind of lab is this? Please specify the City or University.
11. line 144. please specify the equipment manufacturer. From the wavelength, I conclude that the product is yellow and the method is colorimetric. In this case, it is not important that it is a UV-spectrophotometer. Unless the method is fluorescent, in which case excitation and emission data are needed.
12. Subchapter 3.2. should be in raw materials and plant products.
13. Line 256-7 This is a scientific work, so Latin names should be used. Administration of taxonomy, e.g. Linnaeus, is not necessary (if anything, everywhere).
14. Latin names of plants should also be written in italics.
15. The colours of the bars on the charts should be in a different tone than the colour of the product in the test tubes. It is now subconsciously misleading (at first glance). The green control on the graph is blue on the test tube and vice versa.
Author Response
Point 1: In the first paragraph of the Introduction, the authors write about forests in China. And then they smoothly move on to what are supposed to be examples of highly active metabolites of plant origin. Most of them have nothing to do with the forest (e.g. Isatis) and/or China (e.g. Bouvardia). Since this is the main point of connection of the work to the title of the journal as a special issue, it should be corrected. It is really inappropriate to mention tomato or mint here. The selection of species should be considered very carefully. China, as a large country, has a rich flora and you can certainly find native species with valuable properties. A forest is a type of ecosystem and, depending on the geographical latitude has a greater or lesser number of layers and species. However, it is easy to determine whether a given species belongs to such an ecosystem. All names of species should be given in Latin.
Lines 93-4 'forest plants............ orange peel and Fuzhuan brick tea extracts'. Citrus × sinensis and Camelia sinensis or are other species the source of these products and where do they grow in nature? What type of forest is there and where is the source of these species? In line 259 the authors state the same. My question is: Is it true? Are they forest species?
Response 1: Thank you so much for your suggestion. The first paragraph of the Introduction has been rewritten. We do hope that it is now acceptable to the respected reviewer. The details are as follows:
Forests are a significant economic resource in China [1-3], containing rich and widely distributed resources. By 2023, China's forest coverage rate will have reached 24.02%, with 220 million hectares of forest area and 17.76 billion m3 of forest stock. This forest resource includes a rich variety of tea (Camellia sinensis L.) tree species. According to their growth forms and characteristics, tea plants can be divided into arbor-type, small arbor-type, and shrub-type. The arbor-type tea tree varieties are mainly distributed in the South of China, including Guangdong, Guangxi, Fujian and Hainan Provinces. The small arbor-type tea tree varieties are relatively concentrated in Fujian, Guangdong, Guangxi, Hunan and Jiangxi Provinces. Overall, China has a vast area of tea plantations, accounting for about 30% of the world's total tea area. On the other hand, China is also renowned as a treasury of orange resources in the world with a wide variety of orange species, including Citrus reticulata Blanco, Citrus sinensis (L.), Citrus tangerina Tanaka, and so on. The Yangtze River Valley is the largest citrus production region in China, covering an area of more than 200,000 hectares. The main cultivars include oranges, tangerines, and pomelos. This region has an average annual temperature of 17.5-18.5°C and receives about 1,300 millimeters of rain a year. A large number of bioactive compounds with various physiological activities, such as antioxidant activity (catechins and tea polysaccharides extracted from tea), antibacterial activity (flavonoids extracted from orange peel and tea leaves) [4-6], antiviral activity (flavonoids and polyphenols extracted from Isatis tinctoria), antineoplastic activity (paclitaxel extracted from Taxus L), immunoregulatory activity (ginsenoside extracted from Panax ginseng C. A. Mey), antimicrobial activity, lipid-lowering activity, and so on [7-11], could be acquired from different plants. Bouvardin is one of the bacteriostatic agents produced in plant cells when infected by fungus and pathogenic microorganisms or exposed to toxic chemicals and ultraviolet light [12-14]. The antibacterial spectrum of plant extracts is wide, including viruses, fungi and bacteria, which generally inhibit pathogenic microorganism growth and protect probiotics. Due to the high-worth bacteriostatic agent of bouvardin, a novel screening strategy to rapidly obtain various new plant extracts with antimicrobial activity would expand the application of plant forests and enhance plants’ economic value.
Point 2: Subchapter 2.6. Do I understand correctly that the material was bought on the market? Do the authors know anything about the manufacturer? Are they sure of the origins? Are they sure of their species affiliation? From a scientific point of view, this choice of material is not the best.
Line 155 ‘Orange tea leaves and Fuzhuan brick tea’ is probably a mistake. Should it be ‘Orange peel, tea leaves, and Fuzhuan brick tea’?
Response 2: hank you very much. The information has been added. The details are as follows:
Fresh orange (Citrus reticulata Blanco) was purchased from the Greenery Fruit Co. Ltd. (Yiyang Anhua, China). Fresh tea leaves (Camellia sinensis L.) and Fuzhuan brick tea (produced by the same tea leaves) were purchased from Hunan Provincial Baishaxi Industry Co. Ltd. (Yiyang Anhua, China).
Point 3: Was ethanol evaporation a problem? Was the amount of supernatant the same for each raw material?
Response 3: Thank you so much! Ethanol evaporation was not a big problem. Under the condition of being sealed, the sample is extracted with 50% ethanol, so the amount of ethanol volatilization is relatively small. One of the functions of ethanol is to remove large molecules, such as proteins, and improve the solubility of lipid-soluble small molecules. Even if there was little ethanol volatilization, it had little impact on subsequent experiments.
Point 4: What statistical analysis was performed using SPSS? only mean and SD? Are the results statistically significant? There are no markers on the graphs.
Response 4: Thank you very much! The statistical analysis section has been modified. The details are as follows:
All the samples were selected randomly, and the experiments were repeated at least three times. Statistical analyses were performed using SPSS (Statistical Product and Service Solutions) 22.0 statistical software (IBM Corp., Armonk, NY, USA). Values were presented as the mean ± standard deviation. A one-way analysis of variance was performed to evaluate the results. The level of significance was set at p < 0.05.
Point 5: Fig. 3. How was it determined that there was no inhibition? Numerical values, parameters, calculations, or the so-called by-eye, looking at the chart? This should be calculated and quantified.
Response 5: Thank you so much! Figure 3 illustrates the growth curve of E. coli. Notably, the growth curves of these microorganisms remain largely unchanged upon the introduction of the riboswitch fragment. This observation suggests that the introduced riboswitch has a minimal impact on the growth of these microorganisms.
Point 6: In the discussion, the results should be discussed in the context of the chemical composition of the extracts. If the authors do not have their own results, there are many publications on this topic. The possibility of influence through other mechanisms should be taken into account. See line 265 as well.
Response 6: Thank you so much! The discussion has been modified. We do hope that it is now acceptable to the respected reviewer. The details are as follows:
The extract of orange peel demonstrates a capacity to reduce of β-galactosidase activity indicating the presence of compounds may interact with riboswitch. The primary antibacterial components found in orange peel are hesperidin [35], which have a spatial configuration conducive to effective binding with the aptamer section of the riboswitch. In contrast, fresh tea leaf extract has a minimal impact on β-galactosidase activity. However, when processed into Fuzhuan brick tea, the extract's inhibitory effect on the enzyme are significantly enhanced. This phenomenon implies that the fermentation process involved in the production of Fuzhuan brick tea (a fermented dark tea) facilitates the generation of secondary metabolites by microorganisms, which may act as novel antibacterial agents, such as teadenols [36].
Point 7: The names of the bacteria should be given in full when first used.
Response 7: Thank you so much for your suggestion! The full name of English bacteria has been given in Subchapter 2.1. The details are as follows:
Aeromonase hydrophila (A. hydrophila) and Bacillus cereus (B. cereus) were purchased from China General Microbiological Culture Collection Center (Beijing, China). Vibrio cholerae (V. cholerae) O1 biovar eltor strain N16961 genomic DNA was purchased from Shanghai Sangon Biotech Co. Ltd. (Shanghai, China). All plasmid vectors, including pET-24a and pACYC177, were purchased from Thermo Scientific Limited Liability Company (USA). Competent DH5-alpha Escherichia coli (E. coli) was found in our Biolabs (Ocean University of China, Qingdao). The β-galactosidase assays kit was purchased from Shanghai Sangon Biotech Co. Ltd. (Shanghai, China).
Point 8: Line 100-1 What kind of lab is this? Please specify the City or University.
Response 8: Thank you so much! The Synthetic Medicinal Chemistry Lab is at Ocean University of China. The details are as follows:
The C4-HSL was synthesized in the Synthetic Medicinal Chemistry Lab (Ocean University of China, Qingdao).
Point 9: line 144. please specify the equipment manufacturer. From the wavelength, I conclude that the product is yellow and the method is colorimetric. In this case, it is not important that it is a UV-spectrophotometer. Unless the method is fluorescent, in which case excitation and emission data are needed.
Response 9: Thank you so much for your suggestion! The equipment manufacturer has been added. The details are as follows:
The mixture was incubated in a 35 oC water bath for 30 min, and the β-galactosidase activity was measured by an ultraviolet spectrophotometer (UV2600, Shimadzu Corporation, Japan) at 420 nm.
Point 10: Subchapter 3.2. should be in raw materials and plant products.
Response 10: Thank you very much for your suggestion! The sample source has been rewritten in Subchapter 2.6. The orange peel was obtained from a fresh orange. The details are as follows:
Fresh orange (Citrus reticulata Blanco) was purchased from the Greenery Fruit Co. Ltd. (Yiyang Anhua, China). Fresh tea leaves (Camellia sinensis L.) and Fuzhuan brick tea (produced by the same tea leaves) were purchased from Hunan Provincial Baishaxi Industry Co. Ltd. (Yiyang Anhua, China).
Point 11: Line 256-7 This is a scientific work, so Latin names should be used. Administration of taxonomy, e.g. Linnaeus, is not necessary (if anything, everywhere).
Latin names of plants should also be written in italics.
Response 11: Thank you so much for your suggestion! The Latin names of all species in the manuscript have been added, and their italic forms have been checked. The details are as follows:
Introduction: This forest resource includes a rich variety of tea (Camellia sinensis L.) tree species.
On the other hand, China is also renowned as a treasury of orange resources in the world with a wide variety of orange species, including Citrus reticulata Blanco, Citrus sinensis (L.), Citrus tangerina Tanaka, and so on.
A large number of bioactive compounds with various physiological activities, such as antioxidant activity (catechins and tea polysaccharides extracted from tea), antibacterial activity (flavonoids extracted from orange peel and tea leaves) [4-6], antiviral activity (flavonoids and polyphenols extracted from Isatis tinctoria), antineoplastic activity (paclitaxel extracted from Taxus L), immunoregulatory activity (ginsenoside extracted from Panax ginseng C. A. Mey), antimicrobial activity, lipid-lowering activity, and so on [7-11], could be acquired from different plants.
Subchapter 2.1: All the molecular biology reagents were of analytical grade and purchased from Shanghai Sangon Biotech Co. Ltd. (Shanghai, China). Antibiotics were purchased from Sigma-Aldrich (USA). The C4-HSL was synthesized in the Synthetic Medicinal Chemistry Lab (Ocean University of China, Qingdao). DNA oligonucleotides and enzymes were purchased from Sigma-Genosys (USA). Aeromonase hydrophila (A. hydrophila) and Bacillus cereus (B. cereus) were purchased from China General Microbiological Culture Collection Center (Beijing, China). Vibrio cholerae (V. cholerae) O1 biovar eltor strain N16961 genomic DNA was purchased from Shanghai Sangon Biotech Co. Ltd. (Shanghai, China). All plasmid vectors, including pET-24a and pACYC177, were purchased from Thermo Scientific Limited Liability Company (USA). Competent DH5-alpha Escherichia coli (E. coli) was found in our Biolabs (Ocean University of China, Qingdao). The β-galactosidase assays kit was purchased from Shanghai Sangon Biotech Co. Ltd. (Shanghai, China).
Subchapter 2.6: Fresh orange (Citrus reticulata Blanco) was purchased from the Greenery Fruit Co. Ltd. (Yiyang Anhua, China). Fresh tea leaves (Camellia sinensis L.) and Fuzhuan brick tea (produced by the same tea leaves) were purchased from Hunan Provincial Baishaxi Industry Co. Ltd. (Yiyang Anhua, China).
Point 12: The colours of the bars on the charts should be in a different tone than the colour of the product in the test tubes. It is now subconsciously misleading (at first glance). The green control on the graph is blue on the test tube and vice versa.
Response 12: Thank you so much for your suggestion! The colors of the bars in Figures 4, 5, 6 and 7 have been replaced. The details are in our modified manuscript.
Reviewer 2 Report
Comments and Suggestions for Authors
This manuscript reports the construction and use of some transgenic strains containing riboswitches (structured noncoding RNA domains used by many bacteria to monitor the concentrations of their target ligands and regulate gene expression.) that were later used to screen forest plants for antimicrobial agents. The aim of the manuscript is interesting, and the work was apparently well conducted. However, there are several points that need explanation and/or correction. Please see the comments below.
1. The abstract should report better the results obtained, i. e., results should be organized to facilitate their understanding.
2. Keywords: “Plant effective components – I suggest: Plant bioctive compounds; “Antibiotic” – I suggest: antibacterial target; “Riboswitch” – I suggest: noncoding RNA;
3. Introduction section:
- Lines 27-28: “A large number of active compounds have been ...” – please rephrase to: “A large number of bioactive compounds have been ...”
- Lines 28 and 29: “Antioxidants such as extracts from tea (catechins) and tomato (lycopene) have the ...” – please rephrase to “Antioxidants such as catechins and lycopene obtained from tea and tomato extracts respectively have the ...”
- Line 31: “In addition, many substances with various physiological activities...” – please rephrase to: “In addition, many compounds with various physiological activities...”;
- Lines 32 and 33: “antiviral activity (extracts from isatis root), antineoplastic activity (paclitaxel), immunoregulatory activity (extracts from ginseng), antimicrobial activity... – please provide the compounds that are responsible for these activities and not the plants that contain them;
- Line 52: “Members of riboswitches are discovered in the 5’un-...” – Would not be: “Members of riboswitches were discovered in the 5’un-...”;
- Lines 55 and 56: “Riboswitches control a broad range of genes in bacterial species, which involve metabolism...” – please rephrase to: “Riboswitches control a broad range of genes in bacterial species related to metabolism...”
- Line 79: “in vivo” – please use italic form;
4. Material and methods section:
- Please provide a separate section for chemicals employed in the experimental stage. Also, provide in parentheses (Manufacturer, City, Country) for all the chemicals;
- Please provide a separate section for strains and culture conditions. In this section, provide all the scientific names of the bacteria used in full;
Please see also:
- Line 108: “LB medium” – please rephrase to; “Luria-Bertani (LB) medium”;
- Line 129: “Luria-Bertani (LB) broth...” – please rephrase to: “LB broth”;
- Line 132: “The target plasmid (containing CaCl2) was...” – please rephrase to “The target plasmid (containing CaCl2) was...”;
- Line 133: “The mixture was placed in a water bath (42 oC)...” – please rephrase to: “The mixture was placed in a water bath (42 oC)...” and check the entire manuscript;
- Lines 143 and 144: please provide the unit of measurement for the activity of the β-galactosidase;
- Line 145: “... Vc-1, Vc-2, Bc-1, Bc-2, and Ahy1-1 were the same as described...” – please rephrase to ““... Vc-1, Vc-2, Bc-1, Bc-2, and Ahy1-1 were carried out as described...”;
- Line 155: please clarify: in some parts of the manuscript orange peel and tea leaves are mentioned (lines 15 and 16); in others, orange tea leaves and Kuzhuan brick tea (line 155) and even orange peels, tea leaves and Fuzhuan brick tea (lines 259 and 261) - which is correct?
- Line 155: “Orange tea leaves and Fuzhuan brick tea were purchased from the market in Anhua Country...” – please clarify in the manuscript: How do the authors guarantee that the samples were actually orange peels and Kuzhuan brick tea, since there is no correct botanical identification of them? Furthermore, additional sample information must be added to the text, such as manufacturer, country, among others;
- Lines 165 to 168: What statistical tests were performed? The mean tests and significance level must be provided;
5. Results section:
- The quality of the Figures 1, 3, 4A must be improved;
- Figures 4B, 5B, 6B and 7: please provide the unit of measurement of the β-galactosidase activity;
- Figure 5C: please provide the unit of measurement of the growth curve;
- Line 256: “Numerous plants in nature have a bacteriocidal effect...” - This sentence does not reflect the results the authors found, since the samples showed bacteriostatic and not bactericidal effects - please review;
- Line 256 and 257: “... such as Mirabilis jalapa, lemon, and Camellia sinensis...” – please provide all the scientific names in italic form and provide also de scientific name of the lemon;
- Lines 257 and 258: “A compound that significantly reduces gene expression. of c-di-GMP inhibits bacterial growth” - This sentence is is not understandable - please review;
6. Discussion section:
- Lines 305, 312 and 336: “in vivo” – please provide the italic form;
- Lines 334: “The result consists of what was reported by G. Kovacikova. In T. maritima, QS could ...” – please rephrase to: “The result consists of what was reported by Kovacikova [34]. In T. marítima, QS could ...”;
7- Conclusion section: the conclusion presents a summary of the results. It should finalize the findings presented and point out perspectives for the advancement of knowledge in the area studied – please review.
In my final comments, I recommend that the manuscript should be widely reviewed by the authors. The introduction, material and methods, results and conclusion sections must be rephrased to explain more concisely the antibacterial effects.
Comments on the Quality of English LanguageMinor editing of English language required.
Author Response
Point 1: The abstract should report better the results obtained, i. e., results should be organized to facilitate their understanding.
Response 1: Thank you so much! The abstract has been modified. The details are as follows:
Forest plants contain abundant natural products, providing a valuable resource for obtaining compounds with various functional activities, such as antimicrobial, lipid-lowering and immunoregulatory activities. The development of efficient tools for rapidly screening functional natural products from forest plants is essential for human health. In this study, we constructed some transgenic strains (Escherichia coli) containing Ahy1-1 riboswitches that respond to Cyclic di-guanylate (c-di-GMP), serving as a novel bacteriostat target. The Ahy1-1 riboswitches contained the LacZ gene (encoding β-galactosidase) and c-di-GMP aptamer in order to monitor β-galactosidase activity due to changes in c-di-GMP. After co-incubating with extracts from fresh orange peel, fresh tea leaves and Fuzhuan brick tea, the orange peel exhibited significant inhibition of c-di-GMP generation. The extract of tea leaves had a minor influence on the synthesis of c-di-GMP, whereas Fuzhuan brick tea, which is fermented by various microorganisms, inhibited the production of c-di-GMP. Our constructed transgenic strains could be used to screen for antibacterial agents from forest plants. Beyond antibacterial agents, other functional compounds from forest plants could be selected by designing diverse riboswitches.
Point 2: Keywords: “Plant effective components – I suggest: Plant bioctive compounds; “Antibiotic” – I suggest: antibacterial target; “Riboswitch” – I suggest: noncoding RNA;
Response 2: Thank you so much for your suggestion. The keywords have been replaced. The details are as follows:
Keywords: Plant bioctive components; Antibacterial target; Noncoding RNA; Antibiotic resistance; Functional nucleic acid material
Point 3: - Lines 27-28: “A large number of active compounds have been ...” – please rephrase to: “A large number of bioactive compounds have been ...”.
Lines 28 and 29: “Antioxidants such as extracts from tea (catechins) and tomato (lycopene) have the ...” – please rephrase to “Antioxidants such as catechins and lycopene obtained from tea and tomato extracts respectively have the ...”
Line 31: “In addition, many substances with various physiological activities...” – please rephrase to: “In addition, many compounds with various physiological activities...”;
Lines 32 and 33: “antiviral activity (extracts from isatis root), antineoplastic activity (paclitaxel), immunoregulatory activity (extracts from ginseng), antimicrobial activity... – please provide the compounds that are responsible for these activities and not the plants that contain them;
Response 3: Thank you so much for your suggestion. This sentence has been rephrased. The details are as follows:
A large number of bioactive compounds with various physiological activities, such as antioxidant activity (catechins and tea polysaccharides extracted from tea), antibacterial activity (flavonoids extracted from orange peel and tea leaves) [4-6], antiviral activity (flavonoids and polyphenols extracted from Isatis tinctoria), antineoplastic activity (paclitaxel extracted from Taxus L), immunoregulatory activity (ginsenoside extracted from Panax ginseng C. A. Mey), antimicrobial activity, lipid-lowering activity, and so on [7-11], could be acquired from different plants.
Point 4: - Line 52: “Members of riboswitches are discovered in the 5’un-...” – Would not be: “Members of riboswitches were discovered in the 5’un-...”;
Response 4: Thank you so much! The sentence has been confirmed.
Point 5: Lines 55 and 56: “Riboswitches control a broad range of genes in bacterial species, which involve metabolism...” – please rephrase to: “Riboswitches control a broad range of genes in bacterial species related to metabolism...”
Response 5: T Thank you so much for your suggestion. The sentence has been rephrased. The details are as follows:
Riboswitches control a broad range of genes in bacterial species related to metabolism, signal conduction, and uptake of amino acids, nucleotides, cofactors, and metal ions [19].
Point 6: Line 79: “in vivo” – please use italic form; Lines 305, 312 and 336: “in vivo” – please provide the italic form;
Response 6: Thank you so much for your suggestion! All “in vivo” have been italicized.
Point 7: Please provide a separate section for chemicals employed in the experimental stage. Also, provide in parentheses (Manufacturer, City, Country) for all the chemicals;
Please provide a separate section for strains and culture conditions. In this section, provide all the scientific names of the bacteria used in full;
Response 7: Thank you so much! The information on chemicals has been added. The details are as follows:
All the molecular biology reagents were of analytical grade and purchased from Shanghai Sangon Biotech Co. Ltd. (Shanghai, China). Antibiotics were purchased from Sigma-Aldrich (USA). The C4-HSL was synthesized in the Synthetic Medicinal Chemistry Lab (Ocean University of China, Qingdao). DNA oligonucleotides and enzymes were purchased from Sigma-Genosys (USA). Aeromonase hydrophila (A. hydrophila) and Bacillus cereus (B. cereus) were purchased from China General Microbiological Culture Collection Center (Beijing, China). Vibrio cholerae (V. cholerae) O1 biovar eltor strain N16961 genomic DNA was purchased from Shanghai Sangon Biotech Co. Ltd. (Shanghai, China). All plasmid vectors, including pET-24a and pACYC177, were purchased from Thermo Scientific Limited Liability Company (USA). Competent DH5-alpha Escherichia coli (E. coli) was found in our Biolabs (Ocean University of China, Qingdao). The β-galactosidase assays kit was purchased from Shanghai Sangon Biotech Co. Ltd. (Shanghai, China).
Transgenic strains were cultured aerobically to the exponential phase in either Luria-Bertani (LB) medium or kanamycin (50 μg/ml) at 37 °C under continuous shaking (150 rpm).
Point 8: Line 108: “LB medium” – please rephrase to; “Luria-Bertani (LB) medium”;
- Line 129: “Luria-Bertani (LB) broth...” – please rephrase to: “LB broth”;
- Line 132: “The target plasmid (containing CaCl2) was...” – please rephrase to “The target plasmid (containing CaCl2) was...”;
- Line 133: “The mixture was placed in a water bath (42 oC)...” – please rephrase to: “The mixture was placed in a water bath (42 oC)...” and check the entire manuscript;
Response 8: Thank you so much for your suggestion! Those sentences have been rephrased.
Point 9: Lines 143 and 144: please provide the unit of measurement for the activity of the β-galactosidase;
Response 9: Thank you very much! The information has been added. The details are as follows:
The unit of measurement for the activity of β-galactosidase is 10 U/mg.
Point 10: Line 145: “... Vc-1, Vc-2, Bc-1, Bc-2, and Ahy1-1 were the same as described...” – please rephrase to ““... Vc-1, Vc-2, Bc-1, Bc-2, and Ahy1-1 were carried out as described...”;
Response 10: Thank you so much for your suggestion! The sentence has been rephrased. The details are as follows:
The β-galactosidase assays of E. coli strains containing VC-1, VC-2, BC-1, BC-2, and Ahy1-1 were carried out as described above.
Point 11: Line 155: please clarify: in some parts of the manuscript orange peel and tea leaves are mentioned (lines 15 and 16); in others, orange tea leaves and Kuzhuan brick tea (line 155) and even orange peels, tea leaves and Fuzhuan brick tea (lines 259 and 261) - which is correct?
Response 11: Thank you very much. The orange peel, tea leaves and Fuzhuan brick tea are correct. We have also checked the entire manuscript.
Point 12:- Line 155: “Orange tea leaves and Fuzhuan brick tea were purchased from the market in Anhua Country...” – please clarify in the manuscript: How do the authors guarantee that the samples were actually orange peels and Kuzhuan brick tea, since there is no correct botanical identification of them? Furthermore, additional sample information must be added to the text, such as manufacturer, country, among others;
Response 12: Thank you very much. The information on orange peel, tea leaves and Fuzhuan brick tea has been added. The details are as follows:
Fresh orange (Citrus reticulata Blanco) was purchased from the Greenery Fruit Co. Ltd. (Yiyang Anhua, China). Fresh tea leaves (Camellia sinensis L.) and Fuzhuan brick tea (produced by the same tea leaves) were purchased from Hunan Provincial Baishaxi Industry Co. Ltd. (Yiyang Anhua, China).
Point 13: Lines 165 to 168: What statistical tests were performed? The mean tests and significance level must be provided
Response 13: Thank you so much! Information on statistical tests has been added. The details are as follows:
All the samples were selected randomly, and experiments were repeated at least three times. Statistical analyses were performed using SPSS (Statistical Product and Service Solutions) 22.0 statistical software (IBM Corp., Armonk, NY, USA). Values were presented as mean ± standard deviation. A one-way analysis of variance was performed to evaluate the results. The level of significance was set at p < 0.05.
Point 14: The quality of the Figures 1, 3, 4A must be improved;
Figures 4B, 5B, 6B and 7: please provide the unit of measurement of the β-galactosidase activity;
Figure 5C: please provide the unit of measurement of the growth curve;
Response 14: Thank you so much for your suggestion! Figures 1, 3, 4, 5, 6 and 7 have been modified. The details are in our modified manuscript.
Point 15: - Line 256: “Numerous plants in nature have a bacteriocidal effect...” - This sentence does not reflect the results the authors found, since the samples showed bacteriostatic and not bactericidal effects - please review; - Line 256 and 257: “... such as Mirabilis jalapa, lemon, and Camellia sinensis...” – please provide all the scientific names in italic form and provide also de scientific name of the lemon;
Response 15: Thank you very much! The sentence has been deleted.
Point 16: - - - Lines 257 and 258: “A compound that significantly reduces gene expression. of c-di-GMP inhibits bacterial growth” - This sentence is is not understandable - please review;
Response 16: Thank you very much! The sentence “A compound that significantly reduces gene expression of c-di-GMP inhibits bacterial growth” has been replaced by “Low expression of c-di-GMP may slow bacterial growth.”
Point 17: - - - Lines 334: “The result consists of what was reported by G. Kovacikova. In T. maritima, QS could ...” – please rephrase to: “The result consists of what was reported by Kovacikova [34]. In T. marítima, QS could ...”;
Response 17: Thank you so much for your suggestion! The sentence has been rephrased. The details are as follows:
The result consists of what was reported by G. Kovacikova [34]. In T. maritima, QS could adjust the expression of DGC and PDEA genes and then regulate its encoded protein activities.
Point 18: - - 7- Conclusion section: the conclusion presents a summary of the results. It should finalize the findings presented and point out perspectives for the advancement of knowledge in the area studied – please review.
Response 18: Thank you so much for your suggestion! The conclusion has been rewritten. The details are as follows:
In this study, we have developed innovative tools for discovering new antibacterial agents. The synthetic c-di-GMP riboswitches can be utilized to construct inducible knockout lines for essential genes by simply adding the target gene under the c-di-GMP gene and transferring it into the genome to replace the resident gene. We constructed an engineered bacteria containing synthetic c-di-GMP riboswitches to screen for new bacteriostatic agents (c-di-GMP synthesis inhibitors) from forest plants. The introduced riboswitch has minimal impact on the growth of Escherichia coli. Through visualization, it was possible to monitor changes in c-di-GMP in the engineered bacteria treated with various plant extracts. If the engineered bacteria contain two ribose switches, their sensitivity to c-di-GMP can be enhanced. Orange peel, tea leaves and Fuzhuan brick tea were used to screen c-di-GMP synthesis inhibitors. The extracts of orange peel was found to inhibit c-di-GMP synthesis in bacteria. Comparing with fresh tea leaves, processed Fuzhuan brick tea exhibited strong inhibition of c-di-GMP generation. Our engineered bacteria proved to be an efficient tool for rapidly screening c-di-GMP synthesis inhibitors to address antibiotic resistance in pathogenic bacteria from numerous forest plants.
Point 19: - - In my final comments, I recommend that the manuscript should be widely reviewed by the authors. The introduction, material and methods, results and conclusion sections must be rephrased to explain more concisely the antibacterial effects.
Response 19: Thank you so much for your suggestion! The manuscript has been widely reviewed. We do hope that it is now acceptable to the respected reviewer.
Point 20: - - Minor editing of English language required.
Response 20: Thank you so much! The English language has been revised. We do hope that it is now acceptable to the respected reviewer.
Round 2
Reviewer 1 Report
Comments and Suggestions for Authors
Thank you for all the changes. The text is now much better and suitable for publication.
Regards
Reviewer 2 Report
Comments and Suggestions for Authors
All suggestions were considered and corrections were made appropriately.